# Nitric Acid Dissolution of Tennantite, Chalcopyrite and Sphalerite in the Presence of Fe (III) Ions and FeS_2_

**DOI:** 10.3390/ma15041545

**Published:** 2022-02-18

**Authors:** Oleg Dizer, Denis Rogozhnikov, Kirill Karimov, Evgeniy Kuzas, Alexey Suntsov

**Affiliations:** 1Laboratory of Advanced Technologies in Non-Ferrous and Ferrous Metals Raw Materials Processing, Ural Federal University, 620002 Yekaterinburg, Russia; darogozhnikov@yandex.ru (D.R.); kirill_karimov07@mail.ru (K.K.); e.kuzas@yandex.ru (E.K.); 2Institute of Solid State Chemistry, Ural Branch, Russian Academy of Science, 620990 Yekaterinburg, Russia; suntsov@ihim.uran.ru

**Keywords:** chalcopyrite, tennantite, sphalerite, pyrite, nitric acid leaching, optimization, catalytic surface

## Abstract

This paper describes the nitric acid dissolution process of natural minerals such as tennantite, chalcopyrite and sphalerite, with the addition of Fe (III) ions and FeS_2_. These minerals are typical for the ores of the Ural deposits. The effect of temperature, nitric acid concentration, time, additions of Fe (III) ions and FeS_2_ was studied. The highest dissolution degree of sulfide minerals (more than 90%) was observed at a nitric acid concentration of 6 mol/dm^3^, an experiment time of 60 min, a temperature of 80 °C, a concentration of Fe (III) ions of 16.5 g/dm^3^, and an addition of FeS_2_ to the total mass minerals at 1.2:1 ratio. The most significant factors in the break-down of minerals were the nitric acid concentration, the concentration of Fe (III) ions and the amount of FeS_2_. Simultaneous addition of Fe (III) ions and FeS_2_ had the greatest effect on the leaching process. It was also established that FeS_2_ can be an alternative catalytic surface for copper sulfide minerals during nitric acid leaching. This helps to reduce the influence of the passivation layer of elemental sulfur due to the galvanic linkage formed between the minerals, which was confirmed by SEM-EDX.

## 1. Introduction

Copper is a non-ferrous metal that is in high demand on the market. It is produced from monometallic sulfide ores using traditional technologies. The reserves of such raw materials are limited. Therefore, polymetallic ores are given a lot of attention. The ores of Russian deposits are characterized by variety of copper minerals forms and close mutual intergrowth of non-ferrous metal sulfides and iron sulfides. This complicates the production of concentrates with the required extraction of main metals.

Pyrite copper-zinc ores of the Ural region consist of basic sulfides: chalcopyrite (CuFeS_2_), covellite (CuS), chalcocite (Cu_2_S), bornite (Cu_5_FeS_4_), sphalerite (ZnS). Currently, they are often associated with minerals of fahlore ores—tennantite (Cu_12_As_4_S_13_) and tetrahedrite (Cu_12_Sb_4_S_13_) [1]. The presence of arsenic in copper concentrates prevents their processing using traditional technologies. During pyrometallurgical processes, arsenic dust is formed [2,3,4,5,6,7,8,9,10,11,12,13]. It presents a significant danger to humans and the environment [14,15,16]. Antimony negatively affects the quality of copper, reducing its thermal and electrical conductivity.

There have been many studies about this process using various approaches, such as acid leaching [17,18,19,20], ammonia leaching [21], alkaline methods [22,23,24], autoclave oxidation [25,26,27] and bioleaching [28]. They have not achieved industrial implementation for various reasons [29].

The development of an effective method for processing copper polymetallic raw materials with a significant content of fahlore ore minerals is urgent. Nitric acid leaching [30,31,32] makes it possible to achieve the most complete break-down of sulfides and the transfer of valuable metals into solution for their subsequent selective extraction. This method makes it possible to obtain stable and safe commercial products containing arsenic [33,34,35,36] and antimony as components [37].

Minerals of fahlore ores are refractory. The complete dissolution of these minerals and their dissociation products can account for a much larger amount of oxidizer compared to the more widespread copper minerals. This is due to the stepwise dissociation of tennantite and tetrahedrite [38,39]. In this regard, it is advisable to consider the addition of special chemical agents, for example, FeS_2_ and Fe (III) cations. They reduce the nitric acid consumption and accelerate the process. Previously published studies describe the interaction of chalcopyrite, enargite, arsenopyrite and other sulfides with pyrite [40,41,42,43] and iron (III) [44,45,46,47,48,49] in various oxidizing environments. They demonstrate the positive effect of these additives.

The present paper discusses the optimal conditions of nitric acid leaching of natural tennantite, chalcopyrite, and sphalerite in the presence of Fe (III) ions and FeS_2_.

## 2. Materials and Methods

### 2.1. Materials

A mixture of natural sulfide minerals tennantite (Uchalinsky deposit, Sverdlovsk region, Russia), chalcopyrite (Vorontsovsky deposit, North Ural region, Russia), sphalerite (Karabashsky deposit, South Ural region, Russia) was used as the main raw material. The ratio of chalcopyrite, tennantite and sphalerite in the mixture had a ratio of 1:0.36:0.17 by weight. This ratio of minerals is typical for the industrial Cu-As concentrate of the Uchalinsky deposit. The FeS_2_ additive was a natural pyrite mineral (Berezovsky deposit, Sverdlovsk region, Russia). The X-ray diffraction pattern is shown in Figure 1d. The chemical composition of natural minerals is shown in Table 1. All minerals were crushed and sieved. A fraction including 80% of particles with a 20–40 μm size was taken. The granulometric composition of the fraction is shown in Figure 2. Other reagents were analytical grade.

Figure 3 shows scanning electron microscopy (SEM) images of an initial mineral mixture. The chemical composition in certain points, obtained using EDX analysis (Figure 3b), is presented in Table 2. Points 1, 3 and 4 stoichiometrically correspond to chalcopyrite. Point 2 corresponds to tennantite.

Concentrate from the Uchalinsky deposit was used as an industrial copper–arsenic raw material. The particle size of this material was 95% of the −0.074 fraction. The chemical composition is shown in Table 3. The X-ray diffraction pattern of the copper–arsenic concentrate is shown in Figure 4a. Based on the results of the X-ray analysis, the main species of the industrial copper–arsenic raw material were determined: FeS_2_—40%, CuFeS_2_—30%, Cu_12_As_4_S_13_—14%, ZnS—7%, PbS—2%, SiO_2_—2%.

In studies with an industrial concentrate, pyrite from the Berezovsky deposit was used. The chemical composition of pyrite concentrate is presented in Table 4. The X-ray diffraction pattern of pyrite concentrate is shown in Figure 4b.

### 2.2. Apparatuses

Laboratory experiments on nitric acid leaching were carried out on a setup consisting of a borosilicate glass reactor with a jacket Lenz Minni-60 and a volume of 0.5 dm^3^ (Lenz Laborglas GmbH & Co. Wertheim, Germany). The reactor was thermostated using a Huber CC-205B circulating thermostat (HuberKältemaschinenbau AG, Offenburg, Germany). Stirring was carried out using a CatR-100C overhead stirrer (IngenieurbüroCAT, Ballrechten-Dottingen, Germany) at a speed of 350 rpm for efficient mixing of components.

### 2.3. Experiments

The equilibrium composition and the values of the change in the Gibbs energy were calculated using HSC Chemistry Software v. 9.9 (Metso Outotec Finland Oy, Tampere, Finland).

To obtain the optimal parameters of nitric acid leaching, mathematical planning of the experiment was used. StatGraphics 16 was used to construct a second-order orthogonal matrix with five variable parameters: the temperature was 65–95 °C, the nitric acid concentration was 3–8 mol/dm^3^, the concentration of Fe (III) ions was 5–20 g/dm^3^, the mass ratio FeS_2_ to the mixture of sulfides was (0.5–2:1), the time was 15–60 min.

Before the experiment, the solution was heated to a certain temperature. Then, portions of the minerals were added. During the experiment, samples were taken at certain points of time with an automatic dispenser Sartorius Proline (MinebeaIntecAachen GmbH & Co. KG, Aachen, Germany). The final leaching pulp was filtered on a Buchner funnel. The solution was sent for the ICP-MS analysis. The leaching cake was washed with distilled water, and dried at 80 °C until a constant weight. The dried cake was ground on a Pulverisette 6 classic line planetary mill (Fritsch GmbH & Co. KG, Welden Germany), pressed onto a substrate using a hydraulic Vaneox 40t Automatic (Fluxana GmbH & Co. KG., Bedburg-Hau, Germany), and sent for X-ray fluorescence analysis.

### 2.4. Analysis

Chemical analysis of the initial minerals and the solid products was performed using the ARL Advant’X 4200 wave dispersive spectrometer (Thermo Fisher Scientific Inc., Waltham, MA, USA). Phase analysis was carried out on the XRD 7000 Maxima diffractometer (Shimadzu Corp., Tokyo, Japan).

Chemical analysis of the solutions was determined by inductively coupled plasma mass-spectrometry (ICP-MS) using the Elan 9000 instrument (Perkin Elmer Inc., Waltham, MA, USA).

Scanning electron microscopy (SEM) was performed using the JSM-6390LV microscope (JEOL Ltd., Tokyo, Japan) equipped with a module for energy-dispersive X-ray spectroscopy analysis (EDX).

### 2.5. Calculation Method

The dissolution degree of sulfide minerals was calculated using the following procedure:

The mass of dissolved tennantite (mCu12As4S13) was calculated using Formula (1).
(1)$$m_{\mathrm{Cu}12\mathrm{As}4S13}=\frac{\left(C_{\mathrm{As}1}\times V\right)\times M_{\mathrm{Cu}12\mathrm{As}4S13}}{M_{\mathrm{As}}}$$
where C_As1_ is the concentration of arsenic in the leaching solution, determined using ICP-MS, [g/dm^3^]; V is the volume of the leaching solution [dm^3^]; M_Cu12As4S13_ is the molar mass of tennantite, [g/mol]; M_As_ is the molar mass of arsenic in tennantite, [g/mol].

The mass of copper (mCu) in tennantite that passed into the leaching solution was calculated using Formula (2).
(2)$$m_{\mathrm{Cu}}=\frac{m_{\mathrm{Cu}12\mathrm{As}4S13}\times M_{\mathrm{Cu}1}}{M_{\mathrm{Cu}12\mathrm{As}4S13}}$$
where m_Cu12As4S13_ is the mass of dissolved tennantite, [g]; M_Cu12As4S13_ is molar mass of tennantite, [g/mol]; M_Cu1_ is the molar mass of copper in tennantite, [g/mol]

Based on the mass of copper that went into the leaching solution from tennantite, the total mass of dissolved chalcopyrite (m_CuFeS2_) was calculated using Formula (3).
(3)$$m_{\mathrm{CuFeS}2}=\frac{\left(m_{\mathrm{Cu}\left(\mathrm{total}\right)}-m_{\mathrm{Cu}1}\right)\times M_{\mathrm{CuFeS}2}}{2\times M_{\mathrm{Cu}}}$$
where m_Cu(total)_ is the total mass of copper that went into the leaching solution from chalcopyrite and tennantite, [g]; m_Cu1_ is the mass of copper that went into the leaching solution from tennantite, [g]; M_CuFeS2_ is the molar mass of chalcopyrite, [g/mol]; M_Cu_ is the molar mass of copper in chalcopyrite, [g/mol].

The mass of dissolved sphalerite (m_ZnS_) was calculated using Formula (4).
(4)$$m_{\mathrm{ZnS}}=\frac{\left(C_{\mathrm{Zn}}\times V\right)\times M_{\mathrm{ZnS}}}{M_{\mathrm{Zn}}}$$
where C_Zn_ is the concentration of zinc in the leaching solution, determined using ICP-MS, [g/dm^3^]; V is the volume of the leaching solution [dm^3^]; M_ZnS_ is the molar mass of sphalerite, [g/mol]; M_Zn_ is the molar mass of zinc present in sphalerite, [g/mol].

The dissolution degree of tennantite, chalcopyrite, and sphalerite was calculated using Formula (5).
(5)$$\alpha_{\mathrm{MeS}}=\frac{m_{\mathrm{MeS}}}{m_{\mathrm{MeS}}\left(\mathrm{initial}\right)}\times 100$$
where m_MeS_ is the mass of the dissolved mineral, [g]; m_MeS(initial)_ is the initial mass of the mineral in the mixture, [g].

## 3. Results and Discussion

### 3.1. Thermodynamics of Nitric Acid Dissolution of a Mixture of Minerals

To establish the possibility of interaction of sulfide minerals with a nitric acid solution in the presence of Fe (III) ions, the values of the change in the Gibbs energy (ΔG, kJ/mol) were calculated for Equations (6)–(17). The calculation was carried out at an average temperature of 80 °C.
CuFeS_2_ + 16HNO_3_ = FeSO_4_ + CuSO_4_ + 16NO_2_+ 8H_2_O; ΔG^0^_353_ = −1187.2 kJ/mol,(6)
CuFeS_2_ + 10HNO_3_ =Fe(NO_3_)_3_ + Cu(NO_3_)_2_ +2S^0^ + 5NO_2_ + 5H_2_O; ΔG^0^_353_ = −438.198 kJ/mol,(7)
CuFeS_2_ + 22HNO_3_= Fe(NO_3_)_3_ + Cu(NO_3_)_2_ + 2H_2_SO_4_ + 17NO_2_+ 9H_2_O; ΔG^0^_353_ = −1262.5kJ/mol,(8)
FeS_2_ + 18HNO_3_ = Fe(NO_3_)_3_ + 2H_2_SO_4_ + 15NO_2_ + 7H_2_O; ΔG^0^_353_ = −1085.4 kJ/mol,(9)
3FeS_2_ + 14HNO_3_ = 3FeSO_4_ + 3H_2_SO_4_ + 14NO + 4H_2_O; ΔG^0^_353_ = −2763.5 kJ/mol,(10)
Cu_12_As_4_S_13_ + 64HNO_3_ = 12Cu(NO_3_)_2_ + 4H_3_AsO_4_ + 13H_2_SO_4_ + 40NO + 13H_2_O; ΔG^0^_353_ = −1866.5 kJ/mol,(11)
Cu_12_As_4_S_13_ + 38HNO_3_ = 12Cu(NO_3_)_2_ + 4H_3_AsO_4_ + 13S^0^ + 14NO + 13H_2_O; ΔG^0^_353_ = −762.7 kJ/mol,(12)
Cu_12_As_4_S_13_ + 40HNO_3_ = 12CuSO_4_ + 4H_3_AsO_4_ + H_2_SO_4_ + 40NO + 13H_2_O; ΔG^0^_353_= −3647.0 kJ/mol,(13)
ZnS + 8HNO_3_ = ZnSO_4_ + H_2_SO_4_ + 8NO_2_ + 4H_2_O; ΔG^0^_353_= −640.9 kJ/mol,(14)
CuFeS_2_ + 2Fe_2_(SO_4_)_3_ = CuSO_4_ + 5FeSO_4_ + 2S^0^; ΔG^0^_353_ = −65.31 kJ/mol,(15)
ZnS + Fe_2_(SO_4_)_3_ = ZnSO_4_ + 2FeSO_4_ + S^0^; ΔG^0^_353_ = −56.86 kJ/mol(16)
Cu_12_As_4_S_13_ + 13.5Fe_2_(SO_4_)_3_ + 6H_2_O = 12CuSO_4_ + 27FeSO_4_ + 14.5S^0^ + 4H_3_AsO_3_; ΔG^0^_353_ = −28.93 kJ/mol(17)

Based on the results presented above, it can be concluded that the thermodynamic probability of Equations (6)–(17) is quite high.

For the most accurate prediction of the sulfide minerals mixture behavior in the process under study, the graphs of the equilibrium distribution were plotted for their dissolution in nitric acid (Figure 5a) and in a Fe_2_(SO_4_)_3_ solution (Figure 5b).

Pyrite and sphalerite begin to dissolve first when the amount of nitric acid reaches 1 mol (Figure 5a). Chalcopyrite begins to dissolve when the amount of nitric acid reaches 7 mol. For tennantite, this process starts at 9.5 mol of nitric acid. The sequence of sulfide mineral dissolution in a Fe_2_(SO_4_)_3_ solution is similar (Figure 5b).

Therefore, tennantite and chalcopyrite are thermodynamically most resistant under these conditions.

### 3.2. Determination of Optimal Parameters of Nitric Acid Leaching

It was established that for the dissolution of sulfide minerals by more than 90%, a nitric acid concentration of 12 mol/dm^3^ is necessary (Figure 6). High concentrations of nitric acid significantly increase its consumption and increase the cost of the process. Therefore, it is advisable to use additional oxidants and catalysts, such as Fe_2_(SO_4_)_3_ and FeS_2_. This reduces the required concentration and consumption of nitric acid, while maintaining a high degree of sulfide dissolution (not less than 90%).

To determine the influence of FeS_2_ and Fe (III) ions, the experiments were carried out. The parameters of the experiments were as follows: the concentration of nitric acid was 6 mol/dm^3^, the time was 60 min, the temperature was 80 °C, the concentration of Fe (III) ions was 5 g/dm^3^. FeS_2_ was added in mass ratio of 1:1 (to the total mass of sulfide minerals). According to thermodynamic studies, tennantite is the most resistant mineral of the mixture. Therefore, it was chosen as a demonstration of the experimental results (Figure 7).

In the experiment with the simultaneous addition of FeS_2_ and Fe (III) ions, the tennantite dissolution degree increased by 24.5% in 60 min, compared with the experiment without additives. This indicates positive effect of FeS_2_ and Fe (III) ions on the process. The combined use of these additives had the greatest positive effect on the process, compared with their separate use. This effect is possibly explained by the simultaneous catalytic action of FeS_2_ and the oxidative action of Fe (III) ions on the passivating layer of elemental sulfur. The passivating layer forms during the dissolution of minerals. 

To obtain optimal parameters of nitric acid leaching, experiment mathematical planning was used [50,51]. StatGraphics software was used to construct a second-order orthogonal matrix with five variable parameters: the temperature was 65–95 °C, the acid concentration was 3–8 mol/dm^3^, the concentration of Fe (III) ions was 5–20 g/dm^3^, the mass ratio FeS_2_ to the mass of the sulfides mixture was (0.5–2:1), the time was 15–60 min, the liquid-to-solid ratio (L:S) was 6:1. The results of the variance analysis are presented in Table 5.

Figure 8 shows the Pareto diagrams describing the effect of studied parameters on the dissolution process of tennantite, chalcopyrite, and sphalerite.

Considering the data presented in Table 5, it can be concluded that all variable parameters are highly statistically significant for nitric acid leaching of tennantite and chalcopyrite. For sphalerite, the statistically significant parameters are the amount of FeS_2_, Fe (III) ions, and the concentration of nitric acid. The results, presented in Figure 8, confirm these data. FeS_2_ has the greatest influence on the process.

Diagrams of the dependence of the dissolution tennantite, chalcopyrite, and sphalerite on the amount of FeS_2_ and Fe (III) ions at constant values of the concentration of nitric acid (6 mol/dm^3^), time (60 min) and temperature (80 °C) are shown in Figure 9.

The resulting regression Equations (18)–(20) contain the following variables: A—temperature, B—nitric acid concentration, C—concentration of Fe (III) ions, D—amount of FeS_2_, E—time. The adequacy of the selected full quadratic model and regression equations is confirmed by the obtained values of multiple correlation coefficients equal to 0.93 for tennantite, 0.93 for chalcopyrite, and 0.95 for sphalerite.
Cu_12_As_4_S_13_ = −76.52 − 1.4 ∗ A + 8.43 ∗ B + 7.57 ∗ C + 46.72 ∗ D + 1.9 ∗ E + 0.02 ∗ A^2^ − 0.004 ∗ A ∗ B − 0.02 ∗ A ∗ C − 0.06 ∗ A ∗ D − 0.004 ∗ A ∗ E − 0.005 ∗ B^2^ − 0.14 ∗ B ∗ C − 0.05 ∗ B ∗ D − 0.013 ∗ B ∗ E − 0.1 ∗ C^2^ − 0.7 ∗ C ∗ D + 0.0003 ∗ C ∗ E − 6.49 ∗ D^2^ − 0.08 ∗ D ∗ E − 0.009 ∗ E^2^(18)
CuFeS_2_ = −161.17 + 0.26 ∗ A + 18.56 ∗ B + 5.1 ∗ C + 51.81 ∗ D + 2.41 ∗ E + 0.008 ∗ A^2^ − 0.0526667 ∗ A ∗ B − 0.02 ∗ A ∗ C − 0.19 ∗ A ∗ D − 0.005 ∗ A ∗ E − 0.39 ∗ B^2^ − 0.13 ∗ B ∗ C − 1.84 ∗ B ∗ D − 0.012 ∗ B ∗ E − 0.04 ∗ C^2^ − 0.41 ∗ C ∗ D − 0.01 ∗ C ∗ E − 3.1858 ∗ D^2^ − 0.04 ∗ D ∗ E − 0.01 ∗ E^2^(19)
ZnS = −290.24 + 3.57 ∗ A + 25.76 ∗ B + 2.08 ∗ C + 29.78 ∗ D + 3.66 ∗ E − 0.01 ∗ A^2^ − 0.08 ∗ A ∗ B − 0.005 ∗ A ∗ C − 0.19 ∗ A ∗ D − 0.01 ∗ A ∗ E − 0.83 ∗ B^2^ − 0.08 ∗ B ∗ C − 1.09 ∗ B ∗ D − 0.06 ∗ B ∗ E + 0.03 ∗ C^2^ − 0.62 ∗ C ∗ D − 0.014 ∗ C ∗ E + 5.19 ∗ D^2^ − 0.176148 ∗ D ∗ E − 0.02 ∗ E^2^(20)

Therefore, in order to achieve the maximum dissolution degree of tennatite, chalcopyrite, and sphalerite (90% and more), it is necessary to adhere to the values of the Fe (III) concentration, 16.5 mol/dm^3^, weight ratio of FeS_2_ to the mixture of minerals, 1.2:1, concentration of nitric acid, 6 mol/dm^3^, leaching time, 60 min, and temperature, 80 °C.

### 3.3. Comparison of Nitric Acid Leaching of a Natural Mineral Mixture and Industrial Cu-As Concentrate

The optimal parameters of nitric acid leaching of a natural minerals mixture established above were applied to the industrial concentrate of the Uchalinsky deposit. A comparison of the results is shown in Figure 10.

In the period from 0 to 30 min, sulfide minerals in the mixture dissolve more intensively (FeS_2_—99.9%; ZnS—94%; CuFeS_2_—84.8%; Cu_12_As_4_S_13_—75%) than minerals in the industrial concentrate (FeS_2_—99.9%; ZnS—89.2%; CuFeS_2_—75%; Cu_12_As_4_S_13_—66%). This is due to the minerals in the mixture are not associated with each other. The minerals in industrial Cu-As concentrate are disseminated with each other, and the access of nitric acid to them might be partially limited. Despite this, the dissolution degrees of sulfide minerals in a mixture (FeS_2_—99.9%; ZnS—96.7%; CuFeS_2_—94.1%; Cu_12_As_4_S_13_—92.8%) and in an industrial concentrate (FeS_2_—99.9%; ZnS—94.5%; CuFeS_2_—93.2%; Cu_12_As_4_S_13_—91.7%) are almost identical after 60 min.

### 3.4. Characteristics of the Received Cakes

SEM images and EDX mapping for cake obtained at a nitric acid concentration of 6 mol/dm^3^, a time of 60 min, a temperature of 80 °C, and Fe (III) ions concentration of 16.5 g/dm^3^ are shown in Figure 11.

The particles of copper minerals in the nitric acid leaching cake have a nonhomogeneous structure. The green zones in Figure 11f correspond to the distribution of elemental sulfur. The mixture of red and blue zones corresponding to iron and copper refers to chalcopyrite. The surface of unreacted chalcopyrite particles is abundantly covered with elemental sulfur. This can reduce the access of reagents to the reaction surface. The content of sulfur in the cake was 79%. The oxidation of sulfide sulfur in the mixture of minerals to elemental condition reached 56%. The degree of breakdown of pyrite was 88%, tennantite—59%, chalcopyrite—60%, and sphalerite—84%.

SEM images and EDX mapping for cake obtained at a nitric acid concentration of 6 mol/dm^3^, a time of 60 min, a temperature of 80 °C, a Fe (III) ions concentration of 16.5 g/dm^3^, and a mass ratio of FeS_2_ to a mixture of sulfide minerals of 1.2:1 are shown in Figure 12.

According to Figure 12a,b, particles of pyrite and chalcopyrite after nitric acid leaching form conglomerates. They both have smooth and loose, nonhomogeneous surfaces. The green zones in Figure 12f correspond to the elemental sulfur distribution. The red zones are pyrite. The mixture of red and blue zones is chalcopyrite. As in Figure 11, it is noticeable that elemental sulfur covers the surface of chalcopyrite to a greater extent. Its content is minimal on the surface of pyrite.

In the experiment under these conditions, the oxidation degree of sulfide sulfur to elemental sulfur reached 23%, while the cake contained 14%. The degree of break-down of pyrite was 98%, tennantite—93%, chalcopyrite—94%, and sphalerite—99%.

According to the X-ray diffraction patterns (Figure 13), the leaching cakes contain elemental sulfur, chalcopyrite, and tennantite.

Based on the SEM images and EDX mappings presented in Figure 11 and Figure 12, as well as preliminary experiments presented in Figure 7, it can be concluded that during nitric acid leaching, the surfaces of copper minerals are passivated by a film of elemental sulfur. This leads to the limiting of access of nitric acid to the reaction zone. The positive effect of FeS_2_ might be associated with the formation of an electrochemical couple with chalcopyrite and tennantite. In this case, FeS_2_ acts as an alternative surface, as shown in Figure 14. This effect was observed in the studies of joint nitric acid leaching of arsenopyrite and pyrite [52].

Electrochemical nitric acid leaching of chalcopyrite and pyrite in galvanic coupling might be described by the following reaction:CuFeS_2_ + FeS_2_ + NO_3_^−^ = Cu^2+^ + 2Fe^2+^ + SO_4_^2−^ + 4NO_2_ +3S^0^ + 8ē(21)

## 4. Conclusions

In this paper, the study of the nitric acid leaching of a sulfide mineral mixture of pyrite, tennantite, chalcopyrite, and sphalerite, typical for the ores of the Urals deposits, was carried out.

The optimal conditions for nitric acid leaching with the addition of Fe (III) ions and FeS_2_ are: concentration of nitric acid—6 mol/dm^3^; time—60 min; temperature—80 °C; concentration of Fe (III) ions—16.5 mol/dm^3^; mass ratio of amount of FeS_2_ to a mixture of minerals—1.2:1. The dissolution degree of sulfide minerals achieved was more than 90%. Thereby, it was possible to reduce the required concentration of nitric acid from 12 mol/dm^3^ to 6 mol/dm^3^.The most significant factors of the process are: the nitric acid concentration, the concentration of Fe (III) ions, and the amount of FeS_2_. The combination of the addition of Fe (III) ions and FeS_2_ into the process has the greatest effect. The multiple correlation coefficients R^2^ for the obtained regression equations were 0.93 for tennantite, 0.93 for chalcopyrite, and 0.95 for sphalerite, respectively. This indicates the adequacy of the chosen model.During nitric acid leaching of a mixture of sulfide minerals, pyrite can act as an alternative catalytic surface for sulfide copper minerals. Due to the galvanic couple formed between the minerals, it is possible to reduce the influence of the passivating layer.

## Figures and Tables

**Figure 1 materials-15-01545-f001:**
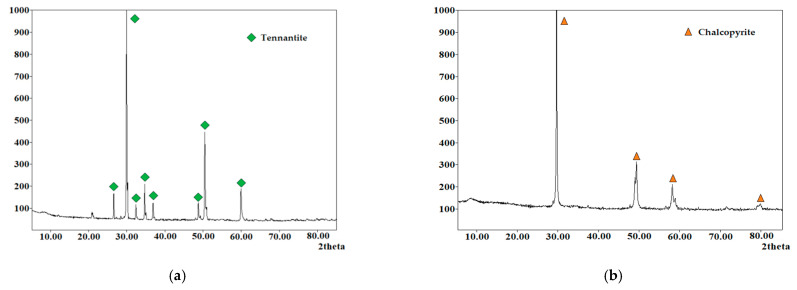
X-ray diffraction pattern of the phase composition: (**a**) tennantite; (**b**) chalcopyrite; (**c**) sphalerite; (**d**) pyrite.

**Figure 2 materials-15-01545-f002:**
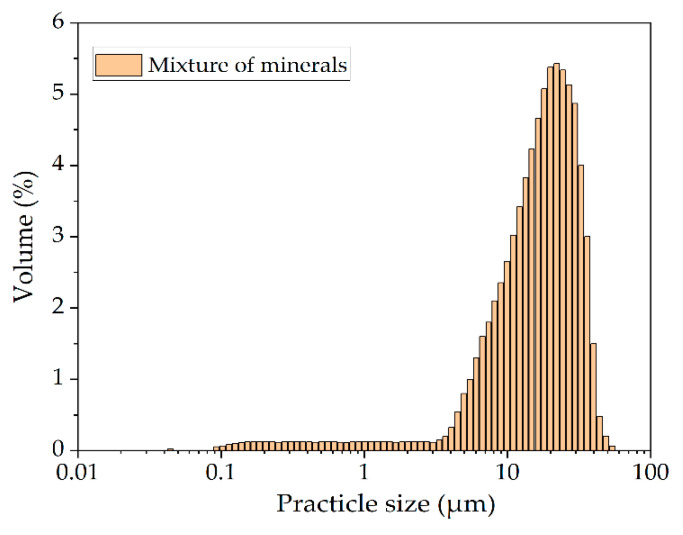
The granulometric composition of sulfide mineral mixture.

**Figure 3 materials-15-01545-f003:**
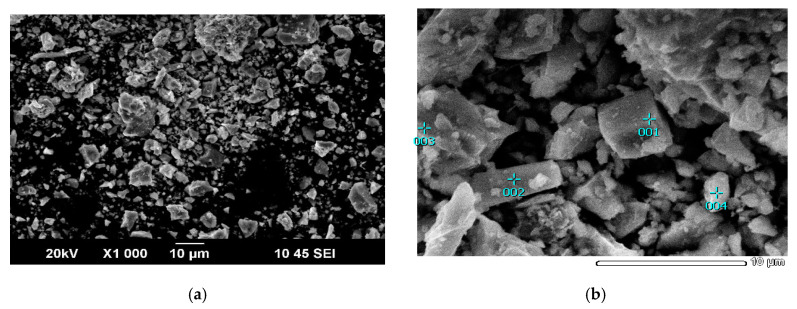
SEM images of the initial mixture (**a**) and some points of determination of the composition (**b**).

**Figure 4 materials-15-01545-f004:**
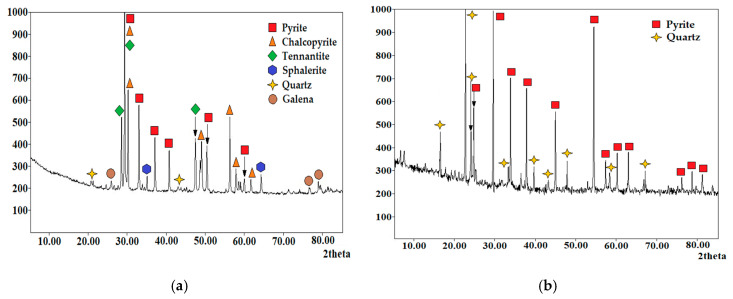
X-ray diffraction pattern of the phase composition: (**a**) Uchalinsky concentrate; (**b**) pyrite concentrate of the Berezovsky deposit.

**Figure 5 materials-15-01545-f005:**
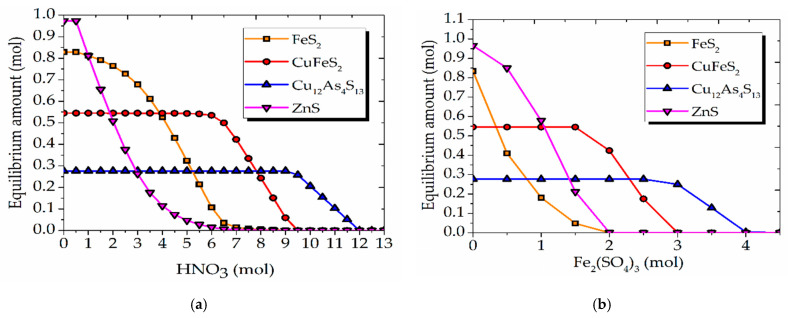
Equilibrium distribution diagrams of pyrite, chalcopyrite, tennantite, and sphalerite in nitric acid solution (**a**) and in Fe_2_(SO_4_)_3_ solution (**b**).

**Figure 6 materials-15-01545-f006:**
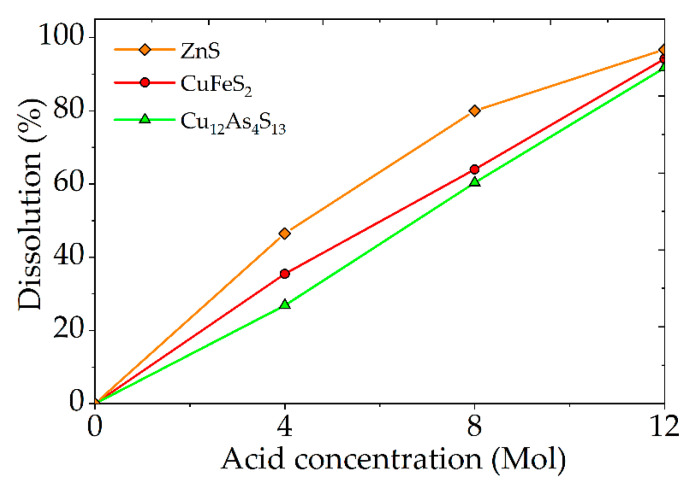
Dependence of the mineral mixture dissolution degree on the nitric acid concentration.

**Figure 7 materials-15-01545-f007:**
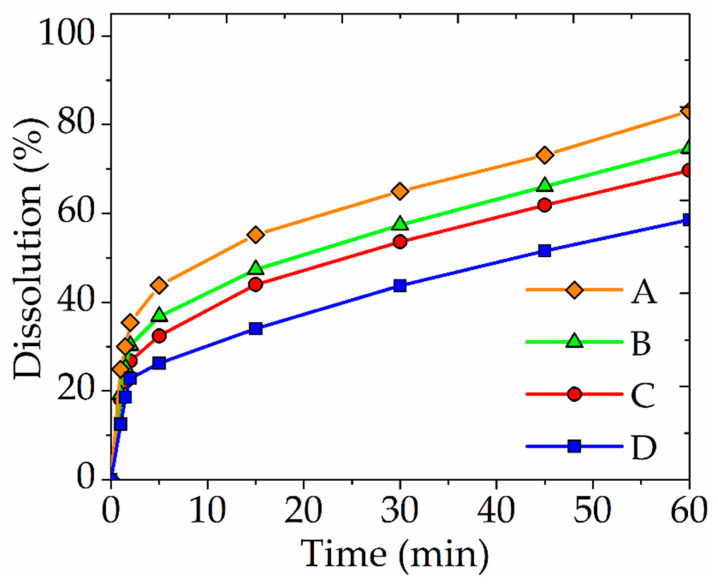
The dependence of the dissolution degree of tennantite in a nitric acid solution on the time with the addition of Fe (III) ions and FeS_2_ (A); with the addition of Fe (III) ions (B); with the addition of FeS_2_ (C); and without additives (D).

**Figure 8 materials-15-01545-f008:**
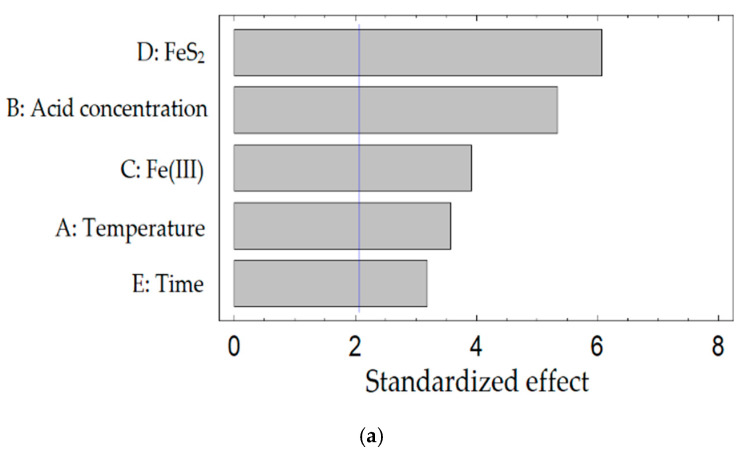
Pareto diagrams for tennantite (**a**), chalcopyrite (**b**), sphalerite (**c**).

**Figure 9 materials-15-01545-f009:**
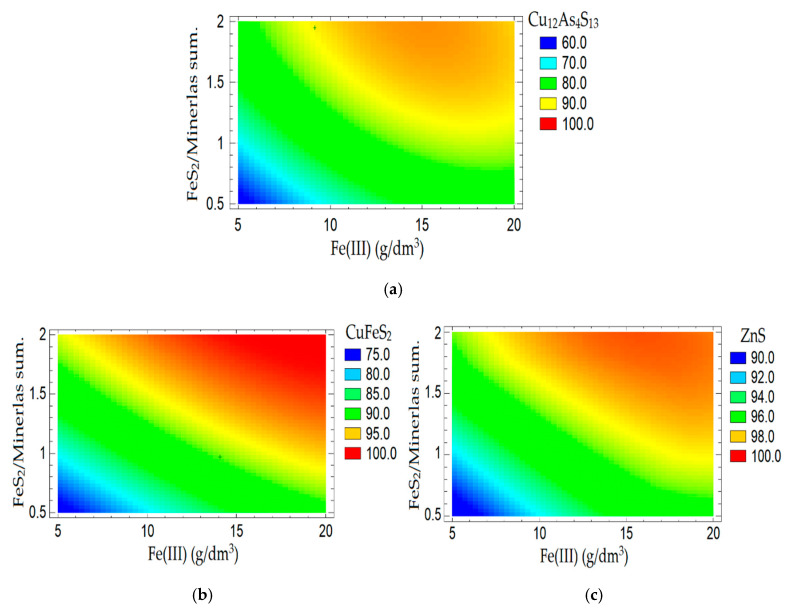
Dependences of the dissolution degree (%) of tennantite (**a**), chalcopyrite (**b**), sphalerite (**c**) on the weight ratio of FeS2 to the mixture of minerals and the concentration of Fe (III) ions, at constant values of the nitric acid concentration, time, and temperature.

**Figure 10 materials-15-01545-f010:**
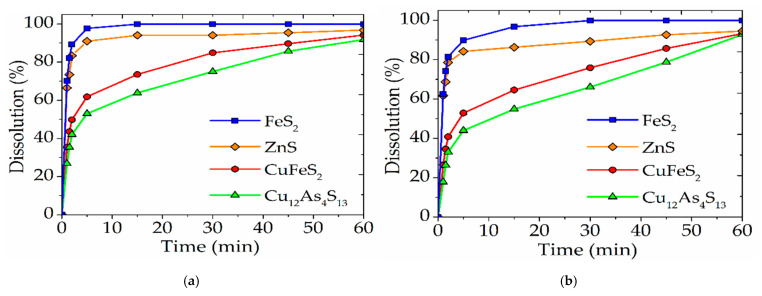
Dependences of the dissolution degree of individual sulfide minerals that are in a mixture with other minerals (**a**), and contained in industrial Cu-As concentrate (**b**), in nitric acid on time.

**Figure 11 materials-15-01545-f011:**
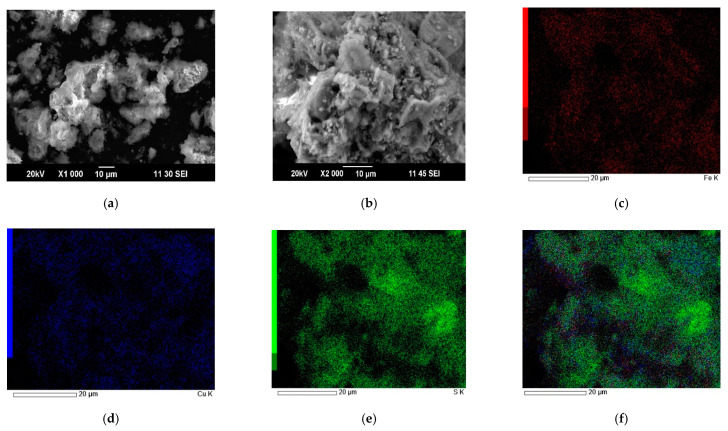
SEM images of leaching cake obtained at a nitric acid concentration of 6 mol/dm^3^, a time of 60 min, a temperature of 80 °C, and Fe (III) ions concentration of 16.5 g/dm^3^. (**a**,**b**) and EDX mapping for iron (**c**), copper (**d**), sulfur (**e**), and combined (**f**).

**Figure 12 materials-15-01545-f012:**
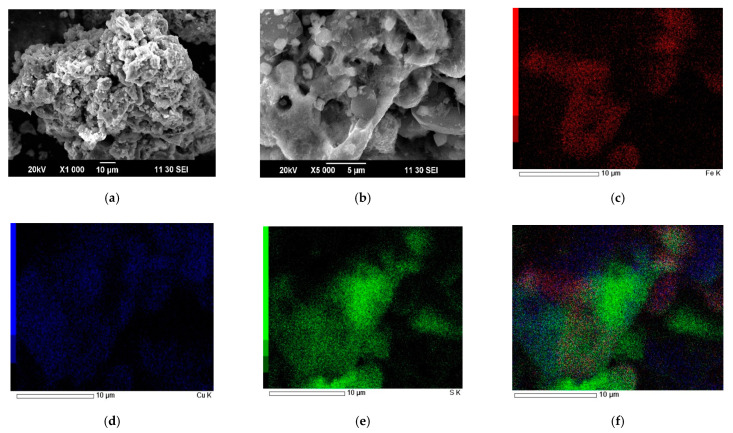
SEM images of leaching cake obtained at a nitric acid concentration of 6 mol/dm^3^, a time of 60 min, a temperature of 80 °C, a Fe (III) ions concentration of 16.5 g/dm^3^, and a mass ratio of FeS_2_ to a mixture of sulfide minerals of 1.2:1. (**a**,**b**) and EDX mapping for iron (**c**), copper (**d**), sulfur (**e**), and combined (**f**).

**Figure 13 materials-15-01545-f013:**
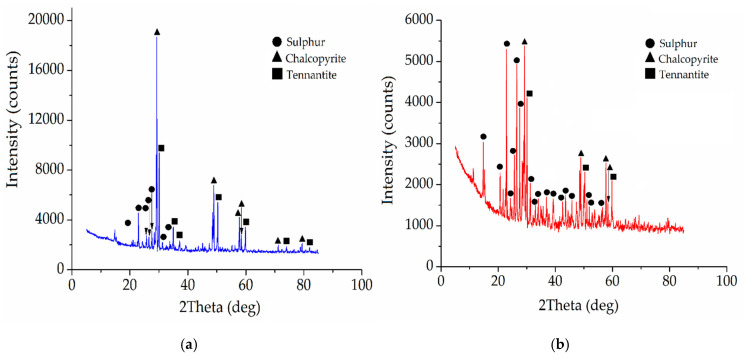
X-ray diffraction patterns of cakes after nitric acid leaching of the minerals mixture with the addition of Fe (III) ions (16 g/dm^3^) (**a**), and with the addition of iron (III) ions (16 g/dm^3^) and weight ratio of FeS_2_ to the minerals mixture—1.2:1 (**b**).

**Figure 14 materials-15-01545-f014:**
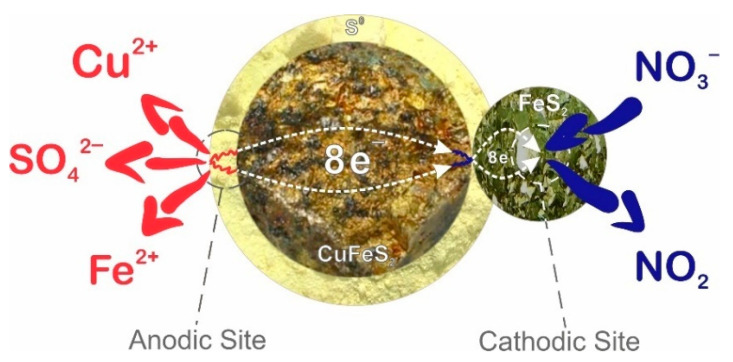
Scheme of electrochemical nitric acid leaching of chalcopyrite and pyrite in galvanic coupling.

**Table 1 materials-15-01545-t001:** The chemical composition of natural minerals, wt. %.

Element/Mineral	S	Fe	Cu	As	Zn	Total
Cu_12_As_4_S_13_	26.8	-	51.7	21.5	-	100.0
CuFeS_2_	36.8	29.8	33.2	-	-	100.0
ZnS	29.9	-	-	-	70.1	100.0
FeS_2_	44.53	55.47	-	-	-	100.0

**Table 2 materials-15-01545-t002:** Results of energy dispersive spectrometry of a mineral mixture, wt. %.

Element	S	Fe	Cu	As	Total
Point 001	32.6	30.4	34.8	1.2	100.0
Point 002	27.1	2.1	51.2	19.6	100.0
Point 003	31.8	29.2	37.2	1.8	100.0
Point 004	36.5	31.6	29.0	2.9	100.0

**Table 3 materials-15-01545-t003:** The chemical composition of Uchalinsky deposit concentrate.

Content, %								
S	Fe	Cu	Zn	As	Pb	Si	Al	Other
39.4	27.9	18.0	4.8	2.9	2.1	0.8	0.5	3.6

**Table 4 materials-15-01545-t004:** The chemical composition of pyrite concentrate.

Content, %				
S	Fe	O_2_	Si	Other
25.8	23.9	23.8	16.4	10.1

**Table 5 materials-15-01545-t005:** The results of the variance analysis.

Source	Sum of Squares	Df	Mean Square	F-Ratio	*p*-Value
Cu_12_As_4_S_13_					
A: Temperature	1896.92	1	1896.92	8.61	0.021
B: Acid concentration	4238.39	1	4238.39	19.25	0.003
C: Fe(III)	2279.08	1	2279.08	10.35	0.014
D: FeS_2_	5471.07	1	5471.07	24.84	0.001
E: Time	1506.64	1	1506.64	6.84	0.034
CuFeS_2_					
A: Temperature	1617.19	1	1617.19	9.01	0.0199
B: Acid concentration	4099.75	1	4099.75	22.83	0.0020
C: Fe(III)	1945.73	1	1945.73	10.84	0.0133
D: FeS_2_	6176.19	1	6176.19	34.39	0.0006
E: Time	1683.8	1	1683.8	9.38	0.0183
ZnS					
A: Temperature	568.548	1	568.548	5.31	0.5486
B: Acid concentration	3537.47	1	3537.47	33.04	0.0007
C: Fe(III)	2880.17	1	2880.17	26.90	0.0013
D: FeS_2_	4405.57	1	4405.57	41.15	0.0004
E: Time	645.106	1	645.106	6.03	0.5938

## Data Availability

The study did not report any data.

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
