# Peer review of "Nitric Acid Dissolution of Tennantite, Chalcopyrite and Sphalerite in the Presence of Fe (III) Ions and FeS_2"

_materials, 2022, doi:10.3390/ma15041545_

Round 1

Reviewer 1 Report

The manuscript entitled "Nitric acid dissolution of tennantite, chalcopyrite, and sphalerite in the presence of Fe(III) ions and FeS2" by Dizer et al., describe the laboratory based experimental works on the acid dissolution techniques of particular sulfide minerals. In this manuscript, authors has described the influences of Fe(III) catalyst on acid-dissolution of those sulfide minerals. In future this technique may be useful for practical uses. However, the overall presentation of the manuscript was not up to the mark; poorly written and needs various changes/ upgradations. Particularly the methodology and results have major lackings. For the same the present draft cannot be recomended for publication in journal like "MATERIALS". 

A few of the major issues are mentioned in the following section-

  • In many places very long statement has been used; and meanings of those statement are not very clear. For example, see the lines 32-35, 63-65, 123-126 etc in the present draft. Such long statements should be splitted into multiple smaller sentences; and that would provide more clarity. A native English -speaker may help in this regard. 
  • In present MS which mainly deals with dissolution technique of metal sulfides; in my opinion, some results (e.g. SEM- EDS) are not very relevant.  
  • The materials/method section needs more expansion. The details of the procedure followed for sample processing and chemical analyses (including ICPMS analyses) should be incorporated. The methodology followed to study the effectiveness of different factors including temperature, concentration of Fe(III), time etc. on the sulfide dissolution should be mentioned in detailed. 
  • The methods for estimation of Gibbs free energy changes for different chemical reactions (as shown in the page 6) should be mentioned in the material/method section. 
  • Replace the Russian fonts with English version. See the formula 3, 5 etc.
  • Ln 13-14 and 20: Fe(III) is ion but how FeS2 has considered as ions ?? 
  •  Ln 36: Cu12As4S13 represents tennantite but not tetrahedrite!! 
  •  Ln 46-47: ......possibilities are limitedly implemented on an in
    dustrial scale for various reasons. -Statement need supportive references. 
  • Ln 53: ....and antimony into commercial products."...-Statement needs references. 
  • In section 2. the description of samples (in subsection 2.2.) should appear before Analytical procedure (of subsection 2.1). 
  • In Tables 1  and 2 the EDS results of different sulfide particles have presented. These results should be interprited  in terms of possible sulfide; otherwise simply proving such composition has no meaning.
  • Ln 146-147: The statement: "Using the formula 2, we calculated the mass of copper (mCu) in tennantite that passed into the solution." should be mentioned after formula 1. 
  • In the title of Table 4, it has mention about "the chemical composition of pyrite concentrate". However, I cannot find any chemical composition in that particular table.   

Author Response

Dear Reviewer. Thank you for your questions and comments. We have changed the text of paper and answered your questions:

Question 1.

In many places very long statement has been used; and meanings of those statement are not very clear. For example, see the lines 32-35, 63-65, 123-126 etc in the present draft. Such long statements should be splitted into multiple smaller sentences; and that would provide more clarity. A native English -speaker may help in this regard.

Answer:

The ores of Russian deposits are characterized by variety of copper minerals forms and close mutual intergrowth of non-ferrous metal sulfides and iron sulfides. This complicates the production of concentrates with the required extraction of main metals. (Lines 29-32)

The present paper discusses the optimal conditions of nitric acid leaching of nat-ural tennantite, chalcopyrite and sphalerite, in the presence of Fe (III) ions and FeS2. (Lines 60-61)

Laboratory experiments on nitric acid leaching were carried out on a setup con-sisting of a borosilicate glass reactor with a jacket Lenz Minni-60 and a volume of 0.5 dm3 (Lenz Laborglas GmbH & Co. Wertheim, Germany). (Lines 112-114)

Question 2.

In present MS which mainly deals with dissolution technique of metal sulfides; in my opinion, some results (e.g. SEM- EDS) are not very relevant.

Answer:

In this article, the results of EDS-mapping prove the assumption of a decrease in the passivation of the surfaces of sulfide minerals with elemental sulfur due to the addition of pyrite particles into the process.

Question 3.

The materials/method section needs more expansion. The details of the procedure followed for sample processing and chemical analyses (including ICPMS analyses) should be incorporated.

Answer:

The solution was sent for the ICP-MS analysis. The leaching cake was washed with dis-tilled water, dried at 80°C until a constant weight. The dried cake ground on a Pulver-isette 6 classic line planetary mill (Fritsch GmbH & Co. KG, Welden Germany), pressed onto a substrate using a hydraulic Vaneox 40t Automatic (Fluxana GmbH & Co. KG., Bedburg-Hau, Germany) and sent for the X-ray fluorescence analysis. (Lines 133-137)

Question 4.

The methodology followed to study the effectiveness of different factors including temperature, concentration of Fe(III), time etc. on the sulfide dissolution should be mentioned in detailed.

Answer:

To obtain the optimal parameters of nitric acid leaching, mathematical planning of the experiment was used. The StatGraphics 16 was used to construct a second-order orthogonal matrix with five variable parameters: the temperature is 65-95 °C, the ni-tric acid concentration is 3-8 mol/dm3, the concentration of Fe (III) ions is 5-20 g/dm3, the mass ratio FeS2 to the mixture of sulfides is (0.5-2:1), the time is 15-60 min. (Lines 124-127)

Question 5.

The methods for estimation of Gibbs free energy changes for different chemical reactions (as shown in the page 6) should be mentioned in the material/method section

Answer:

The equilibrium composition and the values of the change in the Gibbs energy were calculated using the HSC Chemistry Software v. 9.9 (Metso Outotec Finland Oy, Tampere, Finland). (Lines 113-114)

Question 6.

Replace the Russian fonts with English version. See the formula 3, 5 etc.

Answer:

Сorrected

Question 7.

Ln 13-14 and 20: Fe(III) is ion but how FeS2 has considered as ions

Answer:

Сorrected

Question 8.

Cu12As4S13 represents tennantite but not tetrahedrite

Answer:

Сorrected

Question 9.

Ln 46-47: ......possibilities are limitedly implemented on an industrial scale for various reasons. -Statement need supportive references

Answer:

Added

Question 10.

Ln 53: ....and antimony into commercial products."...-Statement needs references.

Answer:

Added

Question 11.

In section 2. the description of samples (in subsection 2.2.) should appear before Analytical procedure (of subsection 2.1)

Answer:

Сorrected

Question 12.

In Tables 1 and 2 the EDS results of different sulfide particles have presented. These results should be interprited in terms of possible sulfide; otherwise simply proving such composition has no meaning.

Answer:

Points 1, 3 and 4 stoichiometrically correspond to chalcopyrite. Point 2 corresponds to tennantite. (Lines 92-93)

Question 13.

Ln 146-147: The statement: "Using the formula 2, we calculated the mass of copper (mCu) in tennantite that passed into the solution." should be mentioned after formula 1.

Answer:

Сorrected

Question 14.

In the title of Table 4, it has mention about "the chemical composition of pyrite concentrate". However, I cannot find any chemical composition in that particular table.

Answer:

Сorrected

Reviewer 2 Report

The manuscript “Nitric acid dissolution of tennantite, chalcopyrite and sphalerite in the presence of Fe (III) ions and FeS2” describes the nitric acid dissolution process of natural minerals such as tennantite, chalcopyrite and sphalerite, which is typical for the ores of the Ural deposits, with the addition of Fe (III) and FeS2 ions into the process. It is defined that the combination of the introduction of Fe (III) and FeS2 ions have the greatest effect on the leach- ing process. Interestingly, it is also determined that pyrite can act as an alternative catalytic surface for copper sulphide minerals during nitric acid leaching of sulphide minerals’ mixture.  This study will also want to explore the mechanism of pyrite reducing the influence of elemental sulfur passivation layer in the process of sulfide dissolution. However, that still requires some revisions before accepting for publication. Please consider the following suggestions:

  1. FeS2exists as a solid, the " FeS2 ions" used in the text is accurate? Please be clear about
  2. In line 114, the “Figure 2a” is Figure 4a? Please be clear about
  3. EDX is only a semi-quantitative analysis, and XPS test could be used tofurther strengthen the analysis of the passive film on the mineral surface
  4. The optimal ratio of FeS2is obtained by statistical analysis? You'd better confirm it through systematic experiments. 
  5. The manuscript needs careful editing by someone with expertise in technical English editing.

Author Response

Dear Reviewer. Thank you for your questions and comments. We have changed the text of paper and answered your questions:

Question 1.

FeS2 exists as a solid, the " FeS2 ions" used in the text is accurate? Please be clear about.

Answer:

Сorrected

Question 2.

In line 114, the “Figure 2a” is Figure 4a? Please be clear about

Answer:

Сorrected

Question 3.

EDX is only a semi-quantitative analysis, and XPS test could be used to further strengthen the analysis of the passive film on the mineral surface

Answer:

Figures 11 and 12 show the distribution of sulfur on the surface of particles after nitric acid leaching using EDX-mapping. The microphotographs show that elemental sulfur covers the surface of chalcopyrite to a greater extent. Its content is minimal on the surface of pyrite. The presence of elemental sulfur is also confirmed by X-ray phase analysis (Fig. 13). We used a similar method of analysis in article – Rogozhnikov, D.; Karimov, K.; Shoppert, A.; Dizer, O.; Naboichenko, S. Kinetics and mechanism of arsenopyrite leaching in nitric acid solutions in the presence of pyrite and Fe(III) ions. Hydrometallurgy 2021. 199, 105525. [https://doi.org/10.1016/j.hydromet.2020.105525].

Question 4.

The optimal ratio of FeS2 is obtained by statistical analysis? You'd better confirm it through systematic experiments

Answer:

The optimal ratio of FeS2 for nitric acid leaching was confirmed in many experiments on pure minerals and commercial Cu-As concentrate.

Question 5.

The manuscript needs careful editing by someone with expertise in technical English editing.

Corrected

Reviewer 3 Report

The content of this manuscript may not fit this journal. Necessary information is missing and the major conclusion is too excessive. The present manuscript should be revised  following problems should be solved by considering the following problems.

  1. The chemical compositions of the three sulfide mineral are necessary.

  1. This study used three sulfides having a high purity. Use of high purity minerals is experimentally correct as the start of research. However, it never practically occurs. I can support the conclusion that the addition of Fe(III) and pyrite is highly effective. However, the optimal values of the other parameters are effective only for the present experimental conditions. Therefore, the first conclusion of the conclusion section on page 15 is not appropriate.

  1. The reference papers or the thermodynamic data base used for the equations 6 – 17 should be presented.

  1. The percentages of the dissolved minerals are shown in Figs. 6, 7, 10. However, these figures do not tell whether the dissolution is stoichiometric or not.

  1. More information is necessary how the calculation in page 9 was performed.

  1. There is a mistake in the caption of Figure 12. The minerals shown in the figure are the residues after leaching test.

  1. The last sentence of the third conclusion on page 15, “it is possible to reduce — elemental sulfur”, is not a conclusion directly derived from the present paper’s result.

Author Response

Dear Reviewer. Thank you for your questions and comments. We have changed the text of paper and answered your questions:

Question 1.

The chemical compositions of the three sulfide mineral are necessary.

Answer:

Added (Line 76)

Question 2.

This study used three sulfides having a high purity. Use of high purity minerals is experimentally correct as the start of research. However, it never practically occurs. I can support the conclusion that the addition of Fe(III) and pyrite is highly effective. However, the optimal values of the other parameters are effective only for the present experimental conditions. Therefore, the first conclusion of the conclusion section on page 15 is not appropriate.

Answer:

The optimal parameters of nitric acid leaching of a natural minerals mixture established above were applied to the industrial concentrate of the Uchalinsky deposit (Lines 267-269).

Question 3.

The reference papers or the thermodynamic data base used for the equations 6 – 17 should be presented

Answer:

The equilibrium composition and the values of the change in the Gibbs energy were calculated using the HSC Chemistry Software v. 9.9 (Metso Outotec Finland Oy, Tampere, Finland). (Lines 121-123)

Question 4.

The percentages of the dissolved minerals are shown in Figs. 6, 7, 10. However, these figures do not tell whether the dissolution is stoichiometric or not

Answer:

Dissolution occurs according to stoichiometry according to the equations given in paragraph 3.1

Question 5.

More information is necessary how the calculation in page 9 was performed

Answer:

Using the design analysis of StatGraphics, the results of the variance analysis were obtained.

Question 6.

There is a mistake in the caption of Figure 12. The minerals shown in the figure are the residues after leaching test.

Answer:

Corrected

Question 7.

The last sentence of the third conclusion on page 15, “it is possible to reduce — elemental sulfur”, is not a conclusion directly derived from the present paper’s result.

Answer:

Corrected

Round 2

Reviewer 1 Report

Polishing of language is still necessary.

Reviewer 3 Report

I think this revision solved the previous problems.